# Synchronized Contrast-Enhanced 4DCT Simulation for Target Volume Delineation in Abdominal SBRT

**DOI:** 10.3390/cancers16234066

**Published:** 2024-12-04

**Authors:** Valeria Faccenda, Denis Panizza, Rita Marina Niespolo, Riccardo Ray Colciago, Giulia Rossano, Lorenzo De Sanctis, Davide Gandola, Davide Ippolito, Stefano Arcangeli, Elena De Ponti

**Affiliations:** 1Medical Physics Department, Fondazione IRCCS San Gerardo dei Tintori, 20900 Monza, Italy; valeria.faccenda@irccs-sangerardo.it (V.F.); elena.deponti@irccs-sangerardo.it (E.D.P.); 2School of Medicine and Surgery, University of Milan Bicocca, 20126 Milan, Italy; riccardo.colciago@unimib.it (R.R.C.); g.rossano1@campus.unimib.it (G.R.); l.desanctis2@campus.unimib.it (L.D.S.); davide.ippolito@unimib.it (D.I.); stefano.arcangeli@unimib.it (S.A.); 3Radiation Oncology Department, Fondazione IRCCS San Gerardo dei Tintori, 20900 Monza, Italy; ritamarina.niespolo@irccs-sangerardo.it; 4Diagnostic Radiology Department, Fondazione IRCCS San Gerardo dei Tintori, 20900 Monza, Italy; davidegiacomo.gandola@irccs-sangerardo.it

**Keywords:** contrast-enhanced imaging, abdominal stereotactic radiotherapy, target volume delineation, respiratory motion management

## Abstract

Stereotactic Body Radiotherapy (SBRT) is emerging as a promising, ablative, non-invasive alternative for treating liver and pancreatic tumors. The high conformality of SBRT, essential to minimize radiation-induced side effects on nearby gastrointestinal organs at risk while ensuring the delivery of high biologically effective doses to the tumor, makes the precise delineation of the target volume even more critical. While four-dimensional computed tomography (4DCT) remains the standard imaging modality for respiratory motion assessment, it often struggles to clearly visualize abdominal tumors due to poor contrast with the surrounding normal tissues in terms of Hounsfield Units (HUs). In this study, we report on our institutional approach to improve target volume delineation and respiratory motion management in abdominal SBRT planning by integrating 4DCT simulation with synchronized intravenous contrast injection.

## 1. Introduction

The use of Stereotactic Body Radiotherapy (SBRT) is emerging as a promising, non-invasive, and ablative treatment option for various abdominal tumors. Evidence supporting SBRT for liver tumors has expanded in the definitive, salvage, and metastatic settings [1,2,3], showing comparable outcomes to those of other modalities such as radiofrequency ablation and transarterial chemoembolization [4,5,6,7,8,9,10]. SBRT has also been integrated into multiple guidelines as a viable option for managing locally advanced or recurrent pancreatic tumors [11,12,13,14,15,16], despite the current lack of high-level prospective evidence [17,18,19].

The high conformality of SBRT, essential for minimizing radiation-induced side effects on the nearby gastrointestinal (GI) organs at risk (OARs) while delivering high biologically effective doses to the tumor, makes the precise delineation of the target volume even more critical. In abdominal sites, tumor positioning is strongly influenced by respiratory motion, with movements exceeding 1–2 cm, especially in the superior–inferior direction, unless specific motion management strategies such as abdominal compression or breath-hold techniques are employed [20,21].

While four-dimensional computed tomography (4DCT) remains the standard imaging modality for assessing respiratory motion, its ability to clearly visualize abdominal tumors is often limited due to poor contrast with the surrounding normal tissues in terms of Hounsfield Units (HUs). To address this limitation, recent guidelines from the European Society for Radiotherapy and Oncology (ESTRO), the former Advisory Committee for Radiation Oncology Practice (ACROP) [22], and the Radiosurgery Society (RSS) [12] recommend using 3D triple-phase protocols for gross tumor volume (GTV) delineation, supplemented with all available types of diagnostic imaging. A non-contrast 4DCT is then used to define the internal target volume (ITV), with an additional 0–5 mm margin for SBRT treatment planning. However, this approach does not fully resolve the challenges of visualizing the tumor during the respiratory cycle. Consequently, margins of ≥2 cm should be applied as if 4DCT were not available [22]; however, such large margins are undesirable in SBRT due to the high doses per fraction involved.

In the past years, dedicated contrast-enhanced 4DCT (ce4DCT) protocols were developed to enhance tumor visualization throughout the breathing cycle [23,24,25,26,27]. Despite their potential advantages, these protocols have not seen widespread adoption across centers, likely due to the technical challenges associated with synchronizing the rapid contrast washout with the prolonged 4D acquisition time, as well as limitations in older and slower CT technologies.

This study presents the technical aspects of our institutional implementation of synchronized ce4DCT for target delineation in abdominal SBRT, utilizing a state-of-the-art helical CT scanner. Additionally, we simulated the commonly used approach of combining triple-phase CT scans with unenhanced 4DCT by using diagnostic CT images, to evaluate the differences in target volume delineation and determine the margins that would be required to account for the target motion.

## 2. Materials and Methods

### 2.1. Baseline 4DCT Procedure

Since March 2022, patients with liver and pancreatic cancer undergoing SBRT at our Radiation Oncology Department, whose lesions were not visible in the baseline scans from previous diagnostic exams, were simulated using a Philips Brilliance CT Big Bore (Koninklijke Philips N.V. Amsterdam, The Netherlands) with two sequential 4DCT scans: one baseline and one contrast-enhanced with synchronized intravenous contrast injection. For immobilization, a 3.2 mm thickness HipFix Thermoplastic mask combined with the customizable Split Leg Vac-Lok vacuum cushion (CQ Medical, Avondale, PA, USA) was employed. Respiratory motion was tracked using integrated pulmonary bellows (Philips Pulmo Toolkit, Philips, Amsterdam, The Netherlands) and, recently, the Sentinel system (C-RAD Positioning AB, Uppsala, Sweden). A standard 4DCT scan with 2 mm slice thickness was acquired, with scan pitch adjustments based on the patient’s respiratory signal, and scan length determined according to the physicians’ requirements. The 4D scan was segmented into ten series of images (phases), each corresponding to a specific point in the patient’s breathing cycle. An untagged volume was also automatically reconstructed from all data of the sinogram.

### 2.2. Personalized Delay Time Calculation

A diagnostic triple-phase CT scan was previously reviewed to identify the optimal contrast phase (arterial, portal, or both) where the tumor was most visible. Contrast timing (*t_phase_*) was initially guided by radiologist recommendations, with *t_phase_* set at 50 s for the arterial phase and at 70 s for the portal phase. Based on this information, a personalized delay time (*t_delay_*) between contrast injection and CT scan was calculated to capture the tumor in the desired phase. Key time points, i.e., the time of the first absolute slice (*t*_0_), the time of the first slice containing the tumor (*t*_1_), and the time of the last slice containing the tumor (*t*_2_), were recorded from the baseline 4DCT in order to obtain the time interval to reach the center of the tumor (*t_scan_ = (t*_1_
*+ t*_2_)/2 − *t*_0_). Even if the tumor was not clearly visible in the baseline scan, anatomical markers and comparisons with the diagnostic CT helped identify its correct positioning. The difference between the theoretical time provided by the radiologists, *t_phase_*, and the time to reach the tumor’s center, *t_scan_*, was used to establish the patient-specific *t_delay_*. Positive *t_delay_* values indicated that the CT scan started after the contrast injection, while negative *t_delay_* values required the CT scan to begin before the contrast injection. A graphical explanation of this workflow is shown in Figure 1.

### 2.3. Contrast-Enhanced 4DCT Procedure

For the ce4DCT, the same 4DCT protocol, including scan length, pitch, slice thickness, and number of phases, was repeated. The iobitridol 350 mgI/mL contrast (Xenetix350, Guerbet, Cedex, France) was administered at the calculated *t_delay_* with a flow rate of 2.5 mL/s and a volume of 1.8–2.0 times the patient’s weight in kg (up to a maximum of 140 mL) using a Medrad Stellant CT Injection System (Bayer Medical Care Inc., Indianola, PA, USA). If *t_delay_* was positive, the process was automated within the scanner to initiate the contrast injection simultaneously with the scanner’s timer countdown. If *t_delay_* was negative, manual synchronization was performed using a stopwatch to trigger the contrast injection at the correct time. The lower-than-typical flow rate (2.5 mL/s vs. 4–5 mL/s) was chosen to slow the contrast washout, ensuring adequate time to capture the tumor in the desired phase.

### 2.4. Contouring and Planning

The two 4DCT acquisitions were fused using rigid registration within Monaco Monte Carlo TPS (Elekta AB, Stockholm, Sweden). Although the patient’s position remained unchanged between scans, internal anatomical changes might occur due to physiological GI movements, making it crucial to align the target area (not the bone anatomy) accurately before contouring. The GTV was delineated on the untagged 4DCT on which the treatment plan was created. This process involved first using the untagged ce4DCT as a secondary reference, followed by incorporating all ten contrast-enhanced respiratory phases to define the ITV. A panel of three experts (a radiologist, a radiation oncologist, and a medical physicist) assessed the lesion visibility on the ce4DCT acquisitions using a straightforward binary scoring system: *visible* or *not visible*. In case of disagreement, a joint review was planned to reach a unanimous consensus. A subclassification of visibility based on the degree of contrast uptake was out of the scope of this research. A 5 mm margin was then added around the ITV to generate the planning target volume (PTV). The GI OARs were contoured on the non-contrast untagged 4DCT. The treatment plans were optimized using the VMAT technique with 1–2 6MV FFF (flattening filter-free) coplanar arcs. The dose prescription ranged from 30 to 35 Gy in 5 fractions for pancreatic cancers and from 50 to 60 Gy in 5 fractions for liver tumors, according to the target volume and nearby OARs.

### 2.5. Data Analysis

Given the reduced flow rate in comparison to that of traditional radiography protocols, the HU values of the aorta were assessed over time on the ce4DCT scans in order to better understand the contrast kinetics in our population. For each untagged ce4DCT slice, a physicist created a circular region of interest (ROI) on the aorta, adjusting its size to match the aortic diameter. A minimum ROI diameter of 5 mm, with at least a 2 mm margin from the aortic edge, was maintained. Mean HU values for each ROI were extracted through MIM software (version 7.1.4, MIM Software Inc., Cleveland, OH, USA) and plotted as a function of time, using the *t_delay_*, the time of the first slice, and the interval between successive slices. By analyzing the actual contrast peak and washout times from prior patients, we were able to determine the optimal *t_phase_* and thus refine the *t_delay_* calculation for subsequent patients. Additionally, patient characteristics such as age, performance status, aortic diameter, contrast flow rate per patient weight, and comorbidities were analyzed using a logistic regression model to explore potential correlations with contrast kinetics and the quality of enhancement. A *p*-value of <0.05 was considered statistically significant. Finally, triple-phase diagnostic CT images were retrieved for each patient and fused with the baseline 4DCT scan using deformable registration focused on the target area in MIM software. A new ITV (ITV2) was delineated following a commonly used approach of integrating the visibility of contrast-enhanced images with the ten phases of the standard 4DCT [22]. The original ITV (ITV1) and ITV2 were then compared by volume, centroid shift, Dice index, Jaccard index, and mean distance agreement (MDA). A *t*-test was also performed to assess the statistical significance of the differences. The margins required to better encompass the ITV1, ensuring that the entire target motion was adequately covered, were calculated using a geometrical method based on the differences between the two volumes in each direction. All statistical analyses were performed using MedCalc^®^ v23.0.2 (MedCalc Software Ltd., Ostend, Belgium).

## 3. Results

Twenty-two patients with liver (*n* = 11) and pancreatic (*n* = 11) cancer were simulated using the described protocol. All ce4DCT scans provided a clear definition of anatomical structures and vessels, and the appropriate *t_delay_* to capture the tumor in the expected *t_phase_* was calculated in all cases. Given that just one liver lesion was barely discernible from the liver parenchyma in the baseline scan, the introduction of a contrast agent substantially improved tumor visibility over the entire breathing cycle in 20 out of 22 cases, according to the three experts’ evaluation, which was unanimous in all cases. The two cases of suboptimal tumor visibility were associated with two hepatocellular carcinomas (HCCs), where, even in the correct phase according to the diagnostic images, the lesion showed minimal contrast uptake. For liver tumors, the late-arterial phase was typically selected, though contrast enhancement varied depending on factors such as lesion type (e.g., HCC vs. metastasis), cirrhosis status, and tumor vascularity. In contrast, for pancreatic tumors, an early portal phase was predominantly chosen to improve the visualization of vascular structures and the pancreatic parenchyma and distinguish the tumor that would not absorb contrast in the same manner. The median differences in HUs between the target and the liver parenchyma were −48 HU for under- and +43 HU for over-enhancement, while that relative to adjacent vessels was −106 HU. Figure 2 shows an example of two different ce4DCT acquisitions with optimal target visibility compared to their correspondent baseline images.

The main parameters of the two sequential 4DCT acquisitions are reported in Table 1. The median ITV delineated on the ten contrast-enhanced phases was 18.9 cc (range, 1.1–108.1). In 8 out of 11 pancreatic cases, a dose of 35 Gy in five fractions was prescribed, while a higher dose of 60 Gy in five fractions was delivered to 5 out of 11 liver cases. The two cases of scarce visibility were treated with 36 Gy in six fractions due to the large target volume and the impossibility to fulfill the GI OAR dose constraints.

The analysis of the aortic HU values revealed that the median times were 54.5 s (range, 39.6–91.4) for the contrast peak and 70.3 s (range, 51.9–106.3) for the washout plateau. The mean ± standard deviation (SD) aortic peak HU value was 292 ± 59, while the mean ± SD plateau HU value was 169 ± 25. In 4 out of 22 cases, anomalous contrast kinetics were observed, with outlying times for both the peak and the plateau. Excluding these outliers, the median values were 53.8 s (range, 45.2–60.6) and 70.0 s (range, 56.0–72.5), respectively. Figure 3 provides an example of one representative standard and one representative anomalous contrast kinetic curve. The logistic regression model did not identify any factors associated with these four instances of abnormal kinetics. The variables analyzed—age, performance status, aortic diameter, contrast flow rate per patient weight in kg, and comorbidities—showed no significant influence (*p* ≥ 0.210) on the contrast distribution within the patients.

For the comparison of the two ITV delineation techniques, the last six patients with liver tumors and six patients with pancreatic tumors were analyzed. The ITV2 values, contoured using the commonly used procedure, were significantly smaller than the ITV1 values obtained through our current approach (paired *t*-test, *p* = 0.045). The mean ± SD percentage difference was −21.2% ± 21.5%. In only one liver case, ITV2 was 13.4% larger than ITV1. The median distance between the centroids of the two volumes was 4.7 mm (range, 0.9–12.6), with 92%, 83%, 58%, and 42% of the ITV2 centroids located more than 1 mm, 2 mm, 4 mm, and 5 mm, respectively, from the corresponding ITV1 centroids. The median overlap between ITV1 and ITV2 was 69% (range, 27–81%) for the Dice index and 53% (range, 16–69%) for the Jaccard index. The median indices for the liver volumes were 46% and 30%, respectively, while the pancreatic volumes showed significantly higher values of 71% (*t*-test, *p* = 0.005) and 55% (*t*-test, *p* = 0.007), respectively. The overall median MDA value was 2.89 mm (range, 1.68–8.43). The median MDA values, separated by tumor type, were 4.80 mm for the liver tumors and 2.46 mm for the pancreatic tumors (*t*-test, *p* = 0.011). Table 2 reports the calculated margins required for ITV2 to encompass ITV1 more effectively. The most frequently mismatched directions across all data were the anterior and the inferior, with mean values exceeding 4 mm, followed by the posterior and the superior, where the mean values were approximately 4 mm. For the liver tumors, the mean margins were nearly or over 5 mm in all directions except for the left axis, while for the pancreatic tumors, only the anterior and the inferior directions exceeded 3 mm. Although a perfect overlap was unattainable due to the irregular shapes of the volumes and the geometric method relying on only six directions, the median indices for ITV2 plus margins relative to ITV1 showed improvement, reaching 79% (range, 69–81%) for the Dice index, 66% (range, 53–69%) for the Jaccard index, and 2.06 mm (range, 1.48–3.03) for MDA, ultimately becoming equivalent for both tumor types (*t*-test, *p* > 0.300).

## 4. Discussion

This study presents an alternative approach for delineating an abdominal target volume throughout all phases of the breathing cycle, within our department. This method enabled a clear visualization of the tumor in all cases except for two, where the liver tumor did not adequately uptake the contrast agent. Factors such as a low contrast agent dosage or the lesion type may account for this issue, which led us to use lower doses and larger margins. In contrast, the pancreatic tumors showed consistent and excellent visualization, achieving a 100% success rate thanks to the optimal contrast enhancement and differentiation from the highlighted parenchyma and vascular structures.

Other studies have explored similar strategies for synchronized ce4DCT techniques. Gupta et al. [25] demonstrated a successful approach for HCCs using the PET/CT technology with cine acquisitions, where two 4DCT scans were performed: one during the arterial phase, and the other during the delayed phase for treatment planning. This approach contrasts with ours, as we first acquire a baseline 4DCT for treatment planning and then a personalized-phase ce4DCT for target delineation. Helou et al. [23] also performed two sequential co-registered 4DCT scans, with the contrast-enhanced arterial phase preceding the non-contrast scan, but they used helical acquisitions, similar to our study. However, they limited the contrast-enhanced scan to a smaller region containing the tumor to reduce the scan time and better synchronize with the contrast injection, excluding in this way the coverage of all OARs. In the study by Beddar et al. [26], 80 patients with common liver tumor types (colorectal liver metastases, cholangiocarcinoma, and HCC) underwent imaging using 150 mL of contrast injected at a flow rate of 5 mL/s. However, the authors only focused on the longer portal venous and delayed phases, using a standard free-breathing 3D scan before the contrast-enhanced one, which did not allow for an assessment of the spatiotemporal extent of the tumor. Finally, Mancosu et al. [27] used a cine acquisition method for pancreatic ductal adenocarcinoma, with parameters tailored to acquire the central part of the pancreas at 50 s after the contrast injection. A common limitation across all these studies is the assumption of uniform contrast peak and transit times for all patients. While consulting with the Radiology Department is recommended to determine the most appropriate parameters, individualizing the contrast timing for each patient would be scientifically more precise. Choi et al. [24] addressed this issue in pancreatic cancer by performing a test bolus contrast injection before the ce4DCT scan, which allowed for an accurate peak enhancement time calculation. Their protocol also included the acquisition of a contrast-enhanced 3DCT with bolus tracking followed by a standard 4DCT acquisition 1–2 weeks before the ce4DCT scan. Although this method increases the procedure accuracy, it adds complexity to it and places additional strain on the patients, who often are already in a vulnerable state.

In this series, we attempted to address the limitation highlighted by Choi et al. [24] by using the aortic enhancement data from previous patients to adjust the *t_phase_* (initially fixed at 50 s for the arterial phase and at 70 s for the portal phase) for subsequent cases. While not fully patient-specific, this approach allowed us to make the process more tailored to the patient population. Although some variability in the contrast arrival times was observed between patients, the median peak time of 54.5 s and washout plateau of 70.3 s aligned with the initial radiology recommendations and were consistent with the findings from several other studies [28,29,30,31]. In only four cases, the contrast distribution deviated from the expected kinetics, leading to a slight phase misalignment (Figure 3 right). These outlier cases were not associated with any of the patient-specific characteristics analyzed, suggesting that further investigation into the cardiac function would have been necessary to better correlate the dynamics of contrast distribution [32,33]. Nonetheless, the target visibility remained unaffected, largely due to the use of the lower contrast flow rate of 2.5 mL/s. This slower contrast distribution extended the time window for optimal tumor visualization, even though the images could appear less sharp compared to diagnostic scans. Indeed, Bae et al. [28] previously showed that a lower injection rate resulted in reduced contrast enhancement, though Gupta et al. [25] found no statistically significant difference between the mean peak HU values in diagnostic contrast 3DCT and ce4DCT scans when using a 3 mL/s injection rate (143.1 HU vs. 134.2 HU, *p* = 0.58).

Our approach is highly dependent on calculating the optimal *t_delay_* for each patient to obtain the window where tumor visualization is maximized, thus avoiding multiple administrations of the intravenous contrast. While a personalized strategy may always seem preferable, achieving some level of standardization within a department could be beneficial for streamlining operations and allowing for the simulation and treatment of more patients. As a next step, we aim to explore the feasibility of standardizing ce4DCT acquisitions within the 54-to-70 s post-injection window. This timing would likely capture the descending phase of contrast enhancement (Figure 3 left), ensuring a high probability of tumor washout visualization. Increased attention will be particularly necessary for hepatic lesions, as their contrast uptake mechanisms can be influenced by multiple factors [23,25,31,34,35,36]. Nevertheless, we believe that investing in a multidisciplinary team to conduct the necessary assessments before simulation CT, including an accurate time calculation between the two 4DCT scans, is worthwhile in modern precision radiotherapy. Furthermore, in the commonly used approach that combines contrast-enhanced 3DCT with non-contrast 4DCT, the physicist must ensure that there are no geometric discrepancies between the two datasets, requiring a careful alignment of up to four images [24,37], while the physician must delineate the tumor across these images as well as ten respiratory phases, where the tumor is often poorly visualized. In comparison, we propose that this method, which only involves two image sets, with a single one used for contouring, is neither excessively time-consuming nor resource-intensive, while offering more reliable tumor visualization and motion assessment.

Our comparative analysis of the two ITV delineation methods underscored the significant influence of patient respiration on target accuracy. A lack of information regarding target motion can lead to significant delineation errors, with a mean ± SD percentage ITV2-ITV1 difference of −21.2% ± 21.5% (*p* = 0.045) and Dice and Jaccard indices as low as 69% and 53%, respectively, on average. Previous studies have demonstrated that single-phase 3DCT is insufficient to capture the full extent of tumor motion [37], with contouring uncertainty being notably higher in unenhanced 4DCT scans [24,38]. Diagnostic triple-phase CT scans may be acquired in varying respiratory phases, and the tumor is not consistently visible across all phases. Godfrey et al. [39] reported significant variability in GTV in pancreas SBRT, primarily due to differences in contrast enhancement across the phases, with overlap ratios ranging from 18% to 75%. Moreover, in our study, the standard 4DCT approach failed to render the tumor visible in all but one case, where the hepatic lesion was faintly distinguishable from the surrounding liver parenchyma. In most cases, motion management was solely reliant on anatomical landmarks. Indeed, we observed that the cases of pancreatic tumors where stents or clips were more commonly present showed better delineation outcomes than the liver tumors, where discrepancies in target volumes were more pronounced. Although the presence of these fiducials can provide a reliable cranio-caudal reference for target delineation even when visibility is poor, these surrogates of motion may not always accurately reflect the actual tumor movement [40,41,42]. This highlights the need for careful consideration when using such markers.

The margins calculated in this study should not be interpreted as recommendations for centers using the approach based on triple-phase CT and unenhanced 4DCT; rather, they emphasize the significant variability that exists among different cases, making standardization impractical. Since most cases showed a centroid mismatch greater than 4 mm, the isotropic margins did not adequately encompass the full range of target motion, resulting in some areas being either too large or too small. Similarly, Cattaneo et al. [38] compared the ITV4D derived from ce4DCT acquisition with a population-based ITV (ITVPBC) defined as the contour of a pancreatic lesion drawn on a single intermediate phase of the ce4DCT, with margins based on centroid motion analysis. They found that ITVPBC was consistently larger than ITV4D when a margin equal to half the maximum centroid excursion was applied to the intermediate-phase GTV, with an average percentage difference of 31.2 ± 17.0%. While increasing margins may ensure that the tumor is targeted, this approach often requires reducing the dose to meet the GI OAR constraints, potentially compromising the ablative effect. Smaller target volumes are more advantageous for delivering higher ablative doses; however, using breath-hold or abdominal compression techniques may be preferable in such cases to limit tumor motion rather than ignoring its full extent. Additionally, one study [24] showed that the tumor volumes delineated on standard 4DCT often overestimated those defined on ce4DCT. This overestimation likely occurred because, in conditions of poor visibility, physicians may tend to contour larger volumes to compensate for uncertainty rather than risk underestimating the tumor’s size.

This study is not devoid of limitations. First, the small sample size limits the generalizability of the findings, and larger studies are needed to validate the technique and assess its impact on clinical outcomes. Second, the volume comparison analysis was influenced by differences in patient positioning between the diagnostic CT and the treatment setup. Variations in patient support—such as the use of a curved table vs. a flat table during diagnostic imaging, with immobilization for treatment—and different respiratory motion management techniques (typically, breath-holding during the fast diagnostic CT) may have affected the results. Additionally, since the diagnostic CT was not acquired for treatment purposes, there may have been a time lapse between the scans, further complicating image fusion. While deformable registration within MIM software should have mitigated these differences, Fukumitsu et al. [43] reported registration accuracy errors ranging from 0.4 to 32.9 mm when using this software, with a noted dependence on the accuracy of rigid registration. Despite these limitations, we believe that this study provides compelling evidence that this approach may effectively reduce systematic errors in target delineation for motion management and improve the accuracy of SBRT planning for abdominal tumors. Moreover, it is readily adaptable for implementation in radiotherapy centers worldwide, regardless of the CT scanner manufacturer, acquisition mode (helical or axial), or respiratory signal monitoring device used. Data from multiple centers could offer valuable insights into the best practices for contrast timing, dosage, protocol standardization, image quality, and patient outcomes.

## 5. Conclusions

The synchronized ce4DCT simulation proved to be feasible and resulted in optimal target delineation over the whole patient’s breathing cycle. This approach only requires minimal resource investment, while significantly mitigating uncertainties in SBRT planning for abdominal tumors by addressing poor tumor visibility and respiratory motion challenges. The combination of triple-phase 3DCT and unenhanced 4DCT introduces great variability in target definition, making isotropic margins ineffective in accurately assessing the target motion without unnecessarily including the surrounding tissues.

## Figures and Tables

**Figure 1 cancers-16-04066-f001:**
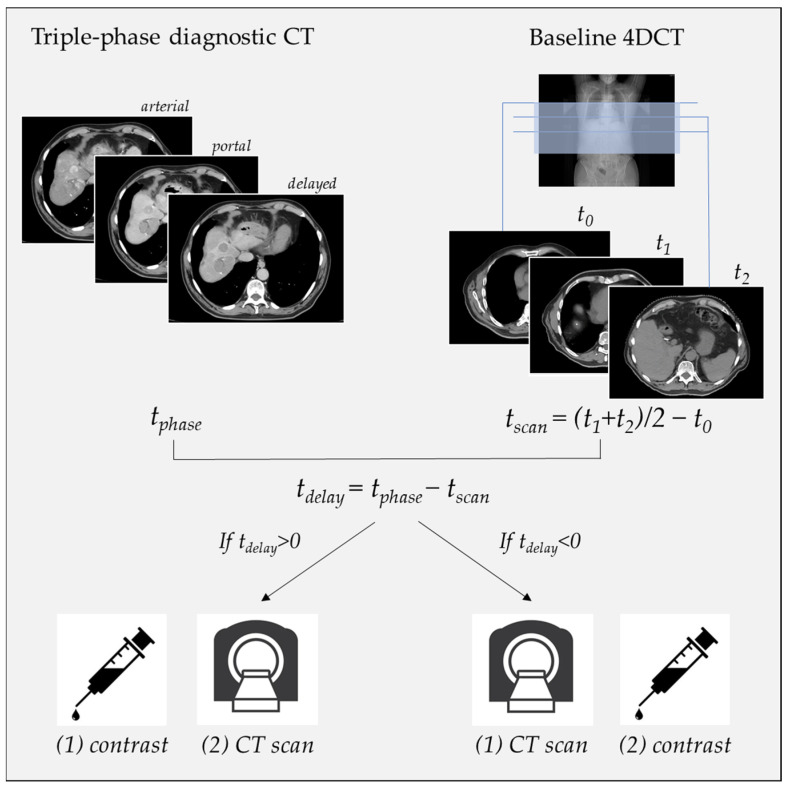
Overview of the implemented workflow for the personalized delay time calculation.

**Figure 2 cancers-16-04066-f002:**
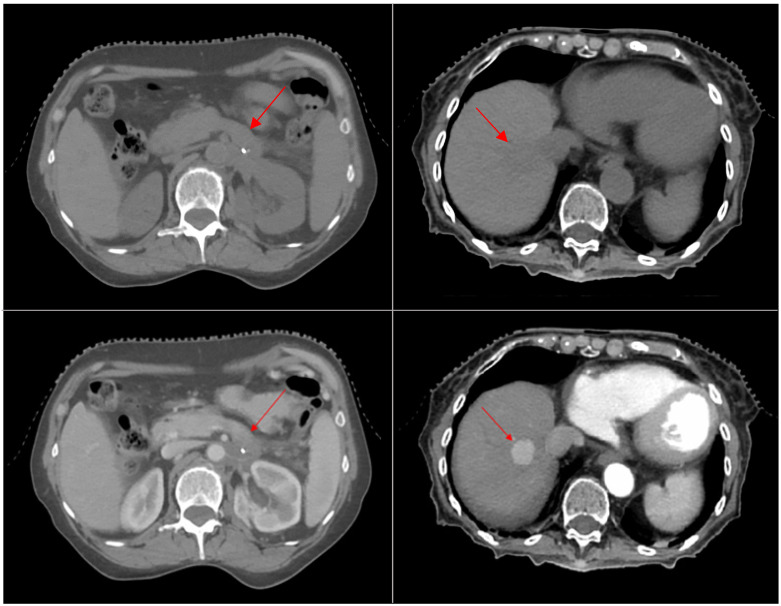
Comparison of baseline (**top**) and contrast-enhanced (**bottom**) 4DCT of a portal phase for a pancreatic lesion (**left**) and of an arterial phase for a liver lesion (**right**). The red arrows indicate the lesion locations.

**Figure 3 cancers-16-04066-f003:**
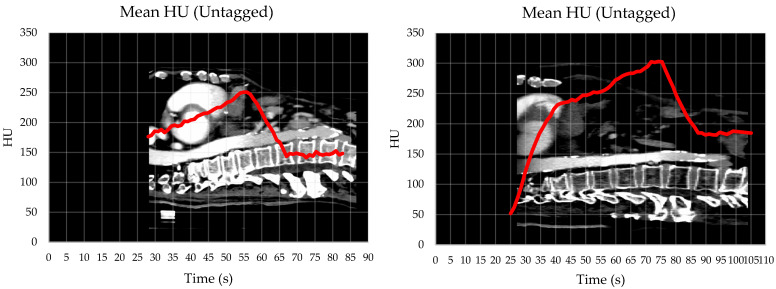
Aortic HU values as a function of time: standard (**left**) and anomalous (**right**) contrast kinetic curves.

**Table 1 cancers-16-04066-t001:** Scan parameters and delay times (*t_delay_*).

Parameter		N. (%)
Median scan length [range]		32.0 cm [23.2–40.8]
Scan pitch	0.041	5 (23%)
	0.059	9 (41%)
	0.08	8 (36%)
Median 4DCT duration [range]		91 s [57–107]
Median *t_delay_* [range]		20 s [−15–37]
Median contrast/kg ratio [range]		1.83 mL/kg [1.33–2.00]

**Table 2 cancers-16-04066-t002:** Calculated margins (mm) required for ITV2 to encompass ITV1 more effectively for each of the twelve analyzed patients. Abbreviations: R = right, L = left, A = anterior, P = posterior, S = superior, I = inferior.

Patient	R	L	A	P	S	I
Liver tumor						
1	8.6	9.3	2.5	8.7	4.0	12.0
2	2.0	2.0	0.0	11.0	7.6	5.9
3	6.4	0.0	5.5	3.5	4.1	6.4
4	0.0	0.0	0.0	0.0	3.6	0.0
5	7.8	5.0	20.7	0.0	4.6	5.0
6	1.9	1.0	0.0	14.5	15.0	2.6
Mean ± SD	4.5 ± 3.6	2.9 ± 3.6	4.8 ± 8.1	6.3 ± 6.0	6.5 ± 4.4	5.3 ± 4.0
Pancreatic tumor						
1	3.1	2.8	5.5	3.1	1.7	7.2
2	0.0	4.4	3.7	3.7	1.4	0.0
3	0.0	0.0	12.0	0.0	0.0	0.0
4	3.7	0.0	0.0	4.0	1.7	3.5
5	0.0	3.5	4.5	0.0	1.3	4.1
6	3.7	4.7	4.5	0.0	1.8	4.5
Mean ± SD	1.8 ± 1.9	2.6 ± 2.1	5.0 ± 3.9	1.8 ± 2.0	1.3 ± 0.7	3.2 ± 2.8
Overall mean ± SD	3.1 ± 3.1	2.7 ± 2.8	4.9 ± 6.1	4.0 ± 4.9	3.9 ± 4.0	4.3 ± 3.5

## Data Availability

Research data are stored in an institutional repository and can be shared upon reasonable request to the corresponding author.

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
