# Peer review of "Synchronized Contrast-Enhanced 4DCT Simulation for Target Volume Delineation in Abdominal SBRT"

_cancers, 2024, doi:10.3390/cancers16234066_

Round 1
Reviewer 1 Report
Comments and Suggestions for Authors
This paper reports about contrast enhanced simulation of target volume definition using 4D CT scan. Only few papers reports on this technical option for liver and pancreatic tumors. The paper is well written and structured. Introduction clearly address the paper topic. Methods are well described and documented with figures. Results are clearly reported. Discussion well focalized the main aspect of this research. I have not fundamental criticism for this manuscript and I suggest the publication in the present form. Minor revision of English language should further improve the quality of the paper.
Reviewer 2 Report
Comments and Suggestions for Authors
I have the following comments:
1) Abstract Results
1a) Lines 35-36. How was tumor visibility determined? Was a semiquantitative scale used? How many readers, and with which degree of experience, were involved in evaluating tumor visibility? Was an inferential statistical test (e.g., Wilcoxon rank test) used to check if the improvement in tumor visibility on ce4DCT acquisitions was statistically significant? This information should be summarized here and reported in full detail in the Results section of the manuscript body.
1b) For all comparisons between continuous variables (i.e. median contrast peak time, washout plateau, liver and pancreatic volumes), p-values should be reported to demonstrate their statistical significance, both here and in the Results section of the manuscript body.
2) Materials and Methods
2a) Line 91. Please replace 'Basal' with 'Baseline'.
2b) Line 125. Please replace 'Xenetix350' with 'iobitridol 350mgI/mL (Xenetix 350, Guerbet, Cedex, France)'.
2c) Lines 132-133. While a lower contrast injection rate can be useful to slow down the contrast washout, it could also lead to a lower enhancement on arterial phase images due to the lower iodine delivery rate, potentially resulting in lower conspicuity of hypervascular lesions and lower overall image quality. What is your comment on this relative to the study findings?
3) Results. See comments 1a and 1b above.
4) Discussion. It would be advisable to briefly discuss to what extent the study findings (obtained on a Philips scanner) could be replicated on CT scanners from other vendors.
5) Overall, the manuscript would benefit from a mild English language revision, ideally by a native English-speaking medical writer or a professional editing service.
Reviewer 3 Report
Comments and Suggestions for Authors
This study provides a method of acquiring contrasted 4DCT images. It has potential for being useful in practice. However, there are a fair number of issues/concerns to be clearly addressed.
Figure 1: I believe tphase and tscan are time intervals (not absolute time). If it is the case, please make it clearer. In specific, explicitly describe tscan = (t1+t2)/2 – t0, I guess.
L127: Indicate the unit of weight (kg, I think) => “the patient’s weight in kg”
L135-139: Per this paragraph, “rigid” image registration depended on the target area more than bony structures, is it correct?
L161-162: Revise – “following the standard approach of” => “following our clinic’s current practice of” if it is correct. If not, => “following a well-reported approach of”
Figure 2: Provide “basal” 4DCT images as well for comparison.
L201-202: Provide how exactly HU values were obtained (e.g., how were the VOIs defined and by whom?). Show examples as well.
L203: Regarding contrast kinetics, show all of 22 curves with 4 outliers in different color & line type from others (e.g., blue vs. red & solid vs. dashed).
L213-215: Revise “standard” as same as in L161-162.
L213-L216: Mean absolute difference was 1.2%, I believe “significantly” here can mislead readers. If I understood correct, make it clearer you mean “statistically significant” but not in absolute amount. Revise every relevant part in the manuscript.
L216-L225: Describe in detail how MDA was obtained. Note, as you mentioned in L135-139, the targets can be placed in different positions with respect to the bony structures. That is, targets in 2 different 4DCT can have 2 different centroids with respect to certain body reference points. In addition, their deformation may not be exactly same either in principle. Thus, MDA may not be meaningful depending on the situation. This issue must be clearly addressed cross the whole manuscript.
Title: “Individually” in the title can mislead readers. Strictly speaking, it is not fully patient specific, please revise the title and some relevant parts in the text.
Round 2
Reviewer 3 Report
Comments and Suggestions for Authors
Most comments/concerns raised have been addressed properly.